# Random Cuts are Optimal for Explainable $k$-Medians

**Konstantin Makarychev**[*]
Northwestern University

**Liren Shan**[*]
TTIC

## Abstract

We show that the RANDOMCOORDINATECUT algorithm gives the optimal competitive ratio for explainable $k$-medians in $\ell_1$. The problem of explainable $k$-medians was introduced by Dasgupta, Frost, Moshkovitz, and Rashtchian in 2020. Several groups of authors independently proposed a simple polynomial-time randomized algorithm for the problem and showed that this algorithm is $O(\log k \log \log k)$ competitive. We provide a tight analysis of the algorithm and prove that its competitive ratio is upper bounded by $2 \ln k + 2$. This bound matches the $\Omega(\log k)$ lower bound by Dasgupta et al (2020).

## 1 Introduction

Machine learning is being increasingly used to make decisions for critical applications, such as healthcare, finance, and public policy. Considering the profound impact of algorithmic decisions on individuals and society, it is essential to understand the underlying logic behind these decisions. In this paper, we explore an explainable $k$-medians clustering algorithm (called RANDOMCOORDINATE-CUT). The algorithm's aim is to cluster data sets and present results in a manner easily understood and visualized by humans.

Clustering is a fundamental task in unsupervised learning. Among many clustering methods, $k$-means, $k$-medians, and $k$-medoids are particularly popular. These are centroid-based methods that choose $k$ centers and assign each data point to the center nearest to it. As a result, each cluster is a Voronoi cell in the Voronoi partition of the space. Since these cells may have a complicated boundary (see Figure 1 for an example of $k$-medians), it is not always easy for humans to comprehend and visualize such clustering.

To address this problem, Dasgupta, Frost, Moshkovitz, and Rashtchian [2020] introduced explainable $k$-means and $k$-medians clustering. They argued that decision trees are easy to understand and interpret. Therefore, in order to make clustering more explainable, we need to use threshold decision trees to define clusters. A threshold decision tree is a binary space partitioning tree with $k$ leaves. Each internal node of the threshold decision tree splits the data into two groups using a threshold cut $(j, \theta)$: on the one side of the cut, we have points $x$ with $x_j \leq \theta$ and, on the other side, points $x$ with $x_j > \theta$. Thus, every node of the tree corresponds to a rectangular region of the space. A decision tree with $k$ leaves partitions data set $X$ into $k$ clusters, $P_1, \ldots, P_k$. See Figure 1 for an example. Dasgupta et al. [2020] suggested that we use the standard $k$-medians and $k$-means objectives to measure the cost of the threshold decision tree. For $k$-medians, the cost of a threshold decision tree $\mathcal{T}$ equals

$$\text{cost}(X, \mathcal{T}) = \sum_{i=1}^{k} \sum_{x \in P_i} \|x - \hat{c}^i\|_1,$$

where $P_1, \ldots, P_k$ is the partitioning of $X$ produced by $\mathcal{T}$; and $\hat{c}^1, \ldots, \hat{c}^k$ are the medians of clusters $P_1, \ldots, P_k$. We denote the $\ell_1$-norm by $\| \cdot \|_1$. Note that each $P_i$ is a rectangular region of the space. Thus, generally speaking, every $x$ is not assigned to the closest center $\hat{c}^1, \ldots, \hat{c}^k$ like in unconstrained $k$-medians or $k$-means.

---

[*]Equal contribution.

37th Conference on Neural Information Processing Systems (NeurIPS 2023).

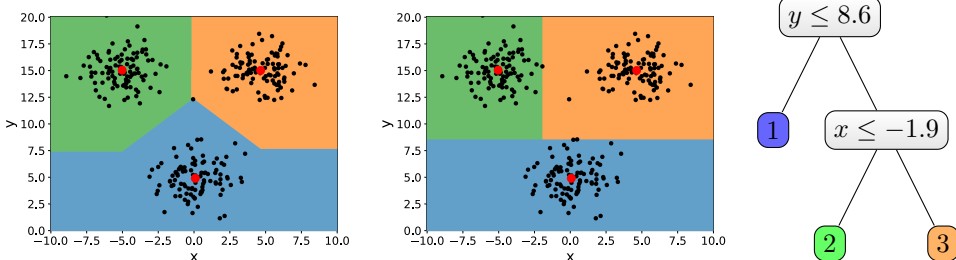

Figure 1: The unconstrained $k$-medians clustering and explainable $k$-medians clustering. The left diagram shows the Voronoi partition of the plane w.r.t. three centers in $\ell_1$ distance. The Voronoi cell for each center consists of all points that are closer (in $\ell_1$ distance) to this center than to any other center (the boundaries between cells are not straight lines because we use the $\ell_1$ distance). The middle diagram shows an explainable partition. The right diagram shows the corresponding decision tree for explainable clustering.

Dasgupta, Frost, Moshkovitz, and Rashtchian [2020] defined the price of explainability as the ratio of the $k$-medians cost of explainable clustering to the optimal cost of unconstrained $k$-medians clustering. They showed that the cost of explainability for $k$-means and $k$-medians (somewhat surprisingly) does not depend on the number of points in the data set $X$ and only depends on $k$. Specifically, they provided a greedy algorithm that given $k$ reference centers $c^1, c^2, \cdots, c^k$ of any unconstrained $k$-medians as input, outputs a threshold decision tree of cost at most $O(k)$ times the cost of original unconstrained $k$-medians with centers $c^1, c^2, \cdots, c^k$. We call such an algorithm $O(k)$ competitive. To get an explainable $k$-medians clustering, we first obtain reference centers $c^1, c^2, \cdots, c^k$ using an off-the-shelf approximation algorithm for $k$-medians and then run an $\alpha$-competitive algorithm for explainable $k$-medians with centers $c^1, c^2, \cdots, c^k$ given as input. This algorithm produces the desired threshold decision tree. Dasgupta et al. [2020] also gave an $O(k^2)$ competitive algorithm for $k$-means and showed $\Omega(\log k)$ lower bounds on the price of explainability for both $k$-medians and $k$-means.

The notion of explainable clustering immediately got a lot of attention in the field (Laber and Murtinho [2021], Makarychev and Shan [2021], Gamlath et al. [2021], Charikar and Hu [2022], Esfandiari et al. [2022]). Particularly, Makarychev and Shan [2021], Esfandiari, Mirrokni, and Narayanan [2022] provided almost optimal algorithms for explainable $k$-medians, and Makarychev and Shan [2021], Esfandiari, Mirrokni, and Narayanan [2022], Gamlath, Jia, Polak, and Svensson [2021] provided almost optimal algorithms for $k$-means. The competitive ratios of these algorithms are $\tilde{O}(\log k)$ for $k$-medians and $\tilde{O}(k)$ for $k$-means.

The algorithms for explainable $k$-medians by Makarychev and Shan [2021], Esfandiari, Mirrokni, and Narayanan [2022], Gamlath, Jia, Polak, and Svensson [2021] are variants of the same simple algorithm, which we call RANDOMCOORDINATECUT. This algorithm receives a set of $k$ reference centers $c^1, \ldots, c^k$ as input and then builds a threshold decision tree with $k$ leaves. It works as follows. It recursively partitions $d$-dimensional space until every cell contains exactly one reference center $c^i$. The algorithm starts with a tree consisting of one node, the root. Initially, all $k$ reference centers are assigned to that root. At every step, the algorithm picks a random threshold cut $(j, \theta)$ and splits centers in every cell using this cut. If this cut does not separate any centers in a cell $u$ (i.e., all centers in $u$ are located on one side of the cut), then the algorithm does not split $u$ into two regions at this step. Finally, for every leaf $u$ of the constructed tree, the unique center that belongs to the cell corresponding to $u$ is assigned to $u$. We provide pseudo-code for this algorithm in Figure 2.

Makarychev and Shan [2021], Esfandiari et al. [2022] showed that the competitive ratio of RANDOMCOORDINATECUT is at most $O(\log k \log \log k)$. That is, for every data set $X$ and set of centers $c^1, \ldots, c^k$,

$$\mathbf{E}[\text{cost}(X, \mathcal{T})] \leq O(\log k \log \log k) \cdot \sum_{x \in X} \min_{c \in \{c^1, \ldots, c^k\}} \|x - c\|_1.$$

Note that the running time of this algorithm is $\tilde{O}(kd)$. Gamlath, Jia, Polak, and Svensson [2021] provided a slightly worse bound of $O(\log^2 k)$ on the competitive ratio of this algorithm. They also

**Input:** a data set $X \subset \mathbb{R}^d$ and set of centers $C = \{c^1, c^2, \ldots, c^k\} \subset \mathbb{R}^d$
**Output:** a threshold tree $\mathcal{T}$

Create tree $\mathcal{T}_0$ containing a root node $r$. Assign $C_r = \{c^1, c^2, \cdots, c^k\}$ to the root. Let $t = 0$. Let $M = \max_{ij} |c_j^i|$.

**while** $\mathcal{T}_n$ contains a leaf with at least two distinct centers **do**
  Pick a coordinate $j$ and threshold $\theta \in (-M, M)$ uniformly at random. Let $\omega_n = (j, \theta)$.
  For every leaf node $u$ in $\mathcal{T}_n$, split the set $C_u$ into two sets:

  $$\textit{Left} = \{c \in C_u : c_j \leq \theta\} \qquad \text{and} \qquad \textit{Right} = \{c \in C_u : c_j > \theta\}.$$

  If both sets are not empty, then create two children of $u$ in tree $\mathcal{T}_t$. The left child corresponds to the subregion of $u$ with $x_j \leq \theta$, and the right child corresponds to the subregion of $u$ with $x_j > \theta$. Assign sets *Left* and *Right* to the left and right child, respectively.

  Denote the updated tree by $\mathcal{T}_{t+1}$.
  Update $t = t + 1$.
**end while**

Figure 2: RANDOMCOORDINATECUT algorithm

conjectured that this algorithm is optimal and its competitive ratio is $O(\log k)$, more specifically, $H_{k-1} + 1$, where $H_k$ is the $k$-th harmonic number. They provided some justification for their conjecture by proving this bound for a very special set of centers and data points (corresponding to the case of completely disjoint sets in our Set Elimination Game).

**Our Results.** In this work, we show that indeed the competitive ratio of RANDOMCOORDINATECUT is at most $2 \ln k + 2$, and, therefore, this algorithm has the optimal competitive ratio which matches the lower bound of Dasgupta, Frost, Moshkovitz, and Rashtchian [2020]. Our analysis is not only tight but also fairly simple. To get our result we define a game, the Set Elimination Game, which was also implicitly analyzed in previous works on this topic. We show that the cost of this game is at most $2 \ln k + 2$.

**Related Work.** The unconstrained $k$-medians clustering has been extensively studied. Charikar, Guha, Tardos, and Shmoys [1999] gave the first constant factor approximation algorithm for the problem in general metric spaces. Li and Svensson [2013] provided a $1 + \sqrt{3} + \varepsilon$ approximation algorithm. Byrka, Pensyl, Rybicki, Srinivasan, and Trinh [2017] improved the approximation factor to $2.675 + \varepsilon$. Cohen-Addad, Esfandiari, Mirrokni, and Narayanan [2022] recently improved the approximation factor to $2.406$ for Euclidean $k$-medians. Megiddo and Supowit [1984] showed that the $k$-medians in $\ell_1$ problem is NP-hard. Cohen-Addad and Lee [2022] showed that it is also NP-hard to approximate $k$-medians in $\ell_1$ within a factor of $1.06$.

As we discuss above, Gamlath, Jia, Polak, and Svensson [2021], Esfandiari, Mirrokni, and Narayanan [2022], Makarychev and Shan [2021], independently proposed the RANDOMCOORDINATECUT algorithm. They also gave an $\tilde{O}(k)$ algorithm for explainable $k$-means and showed a lower bound of $\tilde{\Omega}(k)$ for the problem. Charikar and Hu [2022] provided an $O(k^{1-2/d} \cdot \text{poly}(d, \log k))$ competitive algorithm for explainable $k$-means, whose competitive ratio depends on the dimension $d$ of the instance. For small $d \ll \log k / \log \log k$, their bound is better than $O(k)$. They showed an almost matching $\Omega(k^{1-2/d}/\text{ploy} \log k)$ lower bound for explainable $k$-means. Esfandiari et al. [2022] gave an upper bound of $O(d \log^2 d)$ on the competitive ratio of RANDOMCOORDINATECUT for explainable $k$-medians. This bound is better than $O(\log k)$ for small $d \ll \log k / \log \log k$. Laber and Murtinho [2021] gave $O(d \log k)$ and $O(dk \log k)$ competitive algorithms for explainable $k$-medians and $k$-means, respectively. Frost, Moshkovitz, and Rashtchian [2020] provided some empirical evidence that bi-criteria algorithms for explainable $k$-means (that partition the data set into $(1 + \delta)k$ clusters) can give a much better competitive ratio than $O(k)$. Then, Makarychev and Shan [2022] gave a $\tilde{O}(\frac{1}{\delta} \log^2 k)$ competitive bi-criteria algorithm for explainable $k$-means. Bandyapadhyay, Fomin, Golovach, Lochet, Purohit, and Simonov [2022] provided an algorithm

that computes the optimal explainable $k$-medians and $k$-means clustering in time $n^{2d+O(1)}$ and $(4nd)^{k+O(1)}$, respectively. Laber, Murtinho, and Oliveira [2023] proposed to use shallow decision trees for explainable clustering.

Independently and concurrently with our work, Gupta, Pittu, Svensson, and Yuan [2023] proved a $O(\log k)$ bound on the price of explainability for $k$-medians. They showed that the competitive ratio of RANDOMCOORDINATECUT is $1 + H_{k-1}$, where $H_k$ is the $k$-th harmonic number. Their work answers the open question raised by Gamlath, Jia, Polak, and Svensson [2021]. They also proved a hardness of approximation result for explainable $k$-medians clustering and improved the competitive ratio for explainable $k$-means from $O(k \log k)$ to $O(k \log \log k)$.

## 2  Set Elimination Game

In this section, we define the set elimination game. Consider a discrete finite measure space $(\Omega, \mu)$. In this space, each element $\omega \in \Omega$ has a measure of $\mu(\omega)$, and the measure of every set $S \subseteq \Omega$ equals $\mu(S) = \sum_{\omega \in S} \mu(\omega)$. Let $S_1, S_2, \ldots, S_k \subset \Omega$ be $k$ distinct sets which may overlap with each other. The set elimination game proceeds in a series of rounds. Initially, all sets $S_1, \ldots, S_k$ enter the competition. Formally, they belong to the set of remaining sets $\mathcal{R}_0 = \{S_1, \ldots, S_k\}$. At every round $n$, the host picks a random $\omega_n \in \Omega$ with probability $\Pr(\omega_n = \omega) = \mu(\omega)/\mu(\Omega)$. Then, all sets $S_i$ that contain $\omega_n$ are eliminated from the game unless all remaining sets contain $\omega_n$, in which case, no set gets eliminated. That is, for $n \geq 1$,

$$\mathcal{R}_n = \begin{cases} \mathcal{R}_{n-1} \setminus \{S_i \in \mathcal{R}_{n-1} : \omega_n \in S_i\}, & \text{if for some } S_i \in \mathcal{R}_{n-1}, \omega_n \notin S_i; \\ \mathcal{R}_{n-1}, & \text{otherwise.} \end{cases} \tag{1}$$

The last remaining set is declared the winner. We denote that winner by winner. We say that the cost of the game is the measure of the winning set, $\mu(\text{winner})$.

We remark that $\mathcal{R}_n$ cannot get empty (in which case, the winner would not be defined) because of the "otherwise" clause in the definition (1). We shall always assume that all sets $S_1, \ldots, S_k$ are not only distinct and non-empty but also (a) for every $i$, $\mu(S_i) > 0$, and (b) for all $i$ and $j$, $\mu(S_i \triangle S_j) > 0$ (here, $S_i \triangle S_j$ denotes the symmetric difference of sets $S_i$ and $S_j$). Then, in every game, there is a unique winner with probability 1.

We similarly define the set elimination game for arbitrary finite measure spaces: For an arbitrary finite measure space $(\Omega, \mu)$, element $\omega_n$ is chosen with probability function $\Pr(\omega_n \in S) = \mu(S)/\mu(\Omega)$.

Our main result is the following theorem, which, as we discuss later in Section 2.1, implies that the competitive ratio of the explainable clustering algorithm is $2 \ln k + 2$.

**Theorem 2.1.** *Consider a set elimination game with the finite measure space $(\Omega, \mu)$ and $k$ distinct sets $S_1, S_2, \ldots, S_k$ (as above). The expected cost of the game is at most*

$$\mathbf{E}\big[\mu(\text{winner})\big] \leq (2 \ln k + 2) \cdot \min_{i \in [k]} \mu(S_i).$$

To simplify the exposition, we will prove this theorem for discrete finite measure sets. If $\Omega$ is not a discrete measure space, we first replace it with a quotient space: We say that $\omega' \in \Omega$ and $\omega'' \in \Omega$ are equivalent ($\omega' \sim \omega''$) if they are contained in exactly the same set of sets $S_1, \ldots, S_k$. This equivalence relation partitions $\Omega$ into at most $2^k$ different equivalence classes. We replace $\Omega$ with the quotient space $\Omega/\sim$ whose elements are equivalence classes. In other words, we merge all equivalent $\omega$'s. The measure of a new element $\tilde{\omega}$ equals to the measure of the corresponding equivalence class.

**Organization.** In Section 2.1, we discuss the connection between explainable $k$-medians and set elimination games. We define a set elimination game in a set system $I \subset \{S_1, \ldots, S_k\}$ in Section 2.2. Then, we define the hitting and elimination time in Section 2.3. In Section 3, we first illustrate our proof strategy by showing Theorem 2.1 for the case when the smallest set $S_1$ does not overlap with $S_2, \ldots, S_k$. An important ingredient of our proof is the notion of *surprise sets*, which we discuss in Section 3.1. Finally, we complete the proof of Theorem 2.1 in Section 3.2.

### 2.1  Explainable $k$-Medians via Set Elimination Game

In this section, we show how to use Theorem 2.1 to obtain a bound of $2 \ln k + 2$ on the competitive ratio of the RANDOMCOORDINATECUT algorithm.

**Theorem 2.2.** *The competitive ratio of the* RANDOMCOORDINATECUT *algorithm for Explainable* $k$*-Medians is at most* $2 \ln k + 2$*. That is, for every set of centers* $C = \{c^1, \ldots, c^k\}$ *and data set* $X$*, the algorithm finds a random decision tree* $\mathcal{T}$ *such that*

$$\mathbf{E}[\text{cost}(X, \mathcal{T})] \leq (2 \ln k + 2) \cdot \sum_{x \in X} \min_{c \in \{c^1, \ldots, c^k\}} \|x - c\|_1.$$

The pseudo-code for the RANDOMCOORDINATECUT algorithm is provided in Figure 2.

Theorem 2.2 shows that given any $k$ centers $C = \{c^1, \ldots, c^k\}$, RANDOMCOORDINATECUT finds a decision tree $\mathcal{T}$ with cost at most $2 \ln k + 2$ times the cost of unconstrained $k$-medians with centers $C = \{c^1, \ldots, c^k\}$. By using $k$ centers given by any constant approximation algorithm for $k$-medians, RANDOMCOORDINATECUT finds a decision tree with cost at most $O(\log k)$ times the optimal unconstrained $k$-medians cost. This implies an $O(\log k)$ upper bound on the price of explainability.

*Proof of Theorem 2.2.* Consider an arbitrary data set $X \subset \mathbb{R}^d$ and set of $k$ centers $C \subset \mathbb{R}^d$. We assume that all points in $X$ and all centers in $C$ are in the cube $[-M, M]^d$. The threshold decision tree obtained by the RANDOMCOORDINATECUT algorithm partitions the space into $k$ cells. Each cell contains a single reference center $c^i$. The center $c^i$ is not necessarily optimal for cluster $P_i$ (cluster $P_i$ is the intersection of the data set $X$ and $i$-th cell). However, we will use it as a proxy for the optimal center. In other words, we will upper bound the cost of the threshold decision tree as follows:

$$\text{cost}(X, \mathcal{T}) \equiv \min_{\hat{c}^1, \ldots, \hat{c}^k} \sum_{i=1}^{k} \sum_{x \in P_i} \|x - \hat{c}^i\|_1 \leq \sum_{i=1}^{k} \sum_{x \in P_i} \|x - c^i\|_1.$$

Let $\Omega$ be the set of all coordinate cuts: $\Omega = \{(j, \theta) : j \in [d], \theta \in [-M, M]\}$. We define a measure $\mu$ on $\Omega$ as follows. For every subset $S \subset \Omega$, we set

$$\mu(S) = \sum_{j=1}^{d} \mu_L(\{\theta : (j, \theta) \in S\}),$$

where $\mu_L$ is the Lebesgue measure on $\mathbb{R}$. Thus, we have $\mu(\Omega) = 2dM$, which implies $(\Omega, \mu)$ is a finite measure space.

Consider any data point $x \in X$. Define $k$ sets $S_1, S_2, \ldots, S_k$ for the set elimination game. For every $i \in \{1, \ldots, k\}$, let $S_i$ be the set of all threshold cuts that separate $x$ and center $c^i$, i.e.,

$$S_i = \{(j, \theta) \in \Omega : \text{sign}(x_j - \theta) \neq \text{sign}(c_j^i - \theta)\}.$$

Note that the $\ell_1$ distance from $x$ to center $c^i$ equals the measure of $S_i$: $\|x - c^i\|_1 = \mu(S_i)$. We now examine the set elimination game with sets $S_1, \ldots, S_k$, measure space $(\Omega, \mu)$, and random sequence of draws $\omega_1, \omega_2, \ldots$ (each $\omega_n \in \Omega$ is the threshold cut chosen by the RANDOMCOORDINATECUT algorithm at step $n$). We claim that $S_i$ belongs to $\mathcal{R}_n$ if and only if center $c^i$ lies in the same cell as point $x$ after step $n$ of the algorithm. This is the case for $n = 0$, since $\mathcal{R}_0$ contains all sets $S_1, \ldots, S_k$ and the root of the threshold tree contains all centers $c^1, \ldots, c^k$. Then, whenever we pick cut $\omega_n$, all centers separated from $x$ by $\omega_n$ are removed from the cell of $x$. The only exception from this rule occurs when all centers in that cell lie on the same side of the cut $\omega_n$. That is exactly the same rule as we have for the set elimination game (note that center $c^i$ is separated from $x$ by $\omega_n$ if and only if $\omega_n \in S_i$). Therefore, the same sets $S_i$ remain in the game as center $c^i$ in the cell of $x$ (namely, sets $S_i$ and centers $c^i$ have the same indices).

The RANDOMCOORDINATECUT algorithm stops when all leaves of the decision tree contain exactly one center. At this step, the set elimination game contains one set, $S_i$. This set corresponds to the center $c^i$ assigned to point $x$. The cost of the game $\mu(S_i)$ equals the distance from $x$ to $c^i$. By Theorem 2.1, we have

$$\mathbf{E}[\text{cost}(x, \mathcal{T})] = \mathbf{E}[\mu(\text{winner})] \leq (2 \ln k + 2) \cdot \min_i \mu(S_i) = (2 \ln k + 2) \cdot \min_i \|x - c^i\|_1.$$

We sum this bound over all data points $x$ in $X$ and get the desired result. □

## 2.2 Local Competitions

We now revisit the definition of the set elimination game and define competitions in subsets of $\{S_1, \ldots, S_k\}$. For the rest of the proof, we assume $(\Omega, \mu)$ is a discrete finite measure space. We remind the reader that every set elimination game is determined by an infinite sequence of i.i.d. random variables $\omega_1, \omega_2, \ldots$. In each round $n$, we sample an element $\omega_n$ from $\Omega$ with probability $\Pr(\omega_n = \omega) = \mu(\omega)/\mu(\Omega)$.

**Definition 2.3.** *Consider a finite measure space $(\Omega, \mu)$. Let $I$ be a set of subsets of $\Omega$. We say that $I$ is a valid set system if (a) for every $S \in I$, $\mu(S) > 0$, and (b) for every $S', S'' \in I$, $\mu(S' \triangle S'') > 0$.*

The reader may assume that $\mu(\omega) > 0$ for all $\omega$ in $\Omega$. Then, the definition above says that in a valid set system $I$, all sets are non-empty and distinct.

**Definition 2.4.** *Consider a finite measure space $(\Omega, \mu)$. Let $\omega_1, \omega_2, \ldots$ be i.i.d. random variables as described above and $I$ be a valid set system. We define a set elimination game in $I$. Initially, $\mathcal{R}_0(I) = I$. Then, for every $n \geq 1$,*

$$\mathcal{R}_n(I) = \begin{cases} \mathcal{R}_{n-1}(I) \setminus \{S \in \mathcal{R}_{n-1}(I) : \omega_n \in S\}, & \text{if for some } S' \in \mathcal{R}_{n-1}(I), \omega_n \notin S'; \\ \mathcal{R}_{n-1}(I), & \text{otherwise.} \end{cases} \quad (2)$$

*The winner of the game in $I$, denoted by $\mathrm{winner}(I)$, is the only element remaining, or, formally, the unique element in $\cap_{n \geq 0} \mathcal{R}_n(I)$. If $\cap_{n \geq 0} \mathcal{R}_n(I)$ contains more than one element, then the winner is not defined. The cost of the game is the measure of the winner, $\mu(\mathrm{winner}(I))$.*

We remark that $\cap_{n \geq 0} \mathcal{R}_n(I)$ contains exactly one element with probability 1. Thus, the winner and cost of the game are defined with probability 1.

Consider sets $S_1, \ldots, S_k$ from Theorem 2.1. Denote $K = \{S_1, \ldots, S_k\}$. The definition of the competition among sets $S_1, \ldots, S_k$ (given in the beginning of Section 2) is exactly the same as the definition of competition in $K$. Our goal is to show that $\mathbf{E}[\mu(\mathrm{winner}(K))] \leq 2(\ln k + 1) \cdot \min_{S_i \in K} \mu(S_i)$. In the proof of Theorem 2.1, we will consider competitions in different set systems $I \subseteq K$. We show the following key lemma. We defer the proof of Lemma 2.5 to Appendix A.

**Lemma 2.5.** *Consider a partitioning of the set system $K = \{S_1, \ldots, S_k\}$ into $m$ sets $I_1, \ldots, I_m$. Then, $\mathrm{winner}(K) \in \big\{ \mathrm{winner}(I_1), \ldots, \mathrm{winner}(I_m) \big\}$.*

## 2.3 Set Elimination with Exponential Clock

Consider a set elimination game on sets $S_1, \ldots, S_k$. It is determined by the sequence of random i.i.d. draws $\omega_1, \omega_2, \ldots$. Random variable $\omega_n$ is chosen in round $n$. We assign every round a random time $\tau_n$. Let the time between two consecutive rounds be an exponential random variable with parameter $\mu(\Omega)$. Specifically, let $\Delta\tau_1, \Delta\tau_2, \ldots$ be a sequence of i.i.d. exponential random variables with parameter $\mu(\Omega)$ and each $\tau_n = \tau_{n-1} + \Delta\tau_n = \Delta\tau_1 + \cdots + \Delta\tau_n$. Note that all $\Delta\tau_n$ are positive and $\tau_1, \tau_2, \ldots$ is an increasing sequence with probability 1. The number of draws that occur by time $t$ (i.e., $N_t(\Omega) = |\{n : \tau_n \leq t\}|$) is a Poisson process with parameter $\mu(\Omega)$. We now can think of the set elimination game as follows: The host of the game observes a Poisson process with parameter $\mu(\Omega)$. Whenever the process jumps (at time $\tau_n$), the host picks an element $\omega_n$ in $\Omega$ with probability $\Pr(\omega_n = \omega) = \mu(\omega)/\mu(\Omega)$ and eliminates some sets according to the rules of the game discussed above. Note that by assigning every round some time $\tau_n$, we do not change the game, the winner, and the cost of the game (because the sequence of random draws $\omega_1, \omega_2, \ldots$ remains the same as before). This interpretation of the game allows us to introduce a hitting time $h(S)$ of every subset $S \subset \Omega$ with the following properties: (a) each $h(S)$ is an exponential random variable with rate $\mu(S)$; (b) hitting times of disjoint sets are mutually independent random variables.

**Definition 2.6.** *For every subset $X \subset \Omega$, the hitting time $h(X)$ is the time $\tau_n$ when the first $\omega_n$ is drawn from $X$: $h(X) = \min\{\tau_n : \omega_n \in X\}$. When the set contains one element $\omega$, we will write $h(\omega)$ instead of $h(\{\omega\})$.*

We also define the elimination time of each set $S_i$.

**Definition 2.7.** *Consider any set elimination game with the measure space $(\Omega, \mu)$ and $k$ sets $S_1, S_2, \ldots, S_k$ in $\Omega$. The elimination time $e(S_i)$ of set $S_i$ is the time when set $S_i$ is eliminated from the game, i.e., $e(S_i) = \min\{\tau_n : S_i \notin \mathcal{R}_n(K)\}$. If $S_i$ is the winner, then we let $e(S_i) = \infty$ (because the winner is never eliminated).*

Note that $e(S_i) \geq h(S_i)$. Sometimes, $e(S_i)$ may be equal to $h(S_i)$, but $e(S_i)$ and $h(S_i)$ are not always the same. We now prove that hitting times for disjoint sets are independent. To this end, we *split* the Poisson process $N_t(\Omega) = |\{n : \tau_n \leq t\}|$. Let $N_t(\omega) = |\{n : \tau_n \leq t \text{ and } \omega_n = \omega\}|$. It is easy to see that $N_t(\Omega) = \sum_{\omega \in \Omega} N_t(\omega)$ for every $t$. It is also true that each $N_t(\omega)$ is a Poisson process with parameter $\mu(\omega)$ and all $N_t(\omega)$ (for $\omega \in \Omega$) are mutually independent. This fact follows from the Coloring Theorem (see e.g., Kingman [1992], Coloring Theorem, page 53).

**Theorem 2.8** (Coloring Theorem). *Let $\Pi_t$ be a Poisson process on the real line with rate $\lambda$. We color each event of the Poisson process randomly with one of $M$ colors: The probability that a point receives the $i$-th color is $p_i$. The colors of different points are independent. Let $\Pi_t(i)$ be the number of events of color $i$ in the interval $(0, t]$. Then, $\Pi_t(1), \ldots, \Pi_t(M)$ are independent Poisson processes. The rate of process $\Pi_t(i)$ is $\lambda p_i$.*

**Lemma 2.9.** *For every $\omega \in \Omega$, $h(\omega)$ is an exponential random variable with parameter $\mu(\omega)$, and all random variables $h(\omega)$ (for $\omega \in \Omega$) are mutually independent.*

*Proof.* Observe that $h(\omega) = \min\{t : N_t(\omega) \geq 1\}$. Thus, $h(\omega)$ is an exponential random variable (the time of the first jump of a Poisson process) with rate $\mu(\omega)$. Also, since all $N_t(\omega)$ (for $\omega \in \Omega$) are mutually independent, all $h(\omega)$ are also mutually independent. $\qquad \square$

Note that the set elimination game depends only on the hitting times for elements $\omega$ in $\Omega$. This is the case because it matters only when every $\omega$ is drawn the first time. At that time – the hitting time of $\omega$ – all sets that contain $\omega$ are eliminated unless all remaining sets contain this $\omega$. When the same $\omega$ is drawn again, it does not eliminate any new sets. Also, note that for any set $S \subset \Omega$, the hitting time $h(S) = \min_{\omega \in S} h(\omega)$. Thus, $h(S)$ is an exponential random variable with parameter $\mu(S) = \sum_{\omega \in S} \mu(\omega)$.

## 3 Proof of Main Result

We now present the proof of our main result, Theorem 2.1. We assume without loss of generality that $S_1$ is the smallest set i.e., $\mu(S_1) \leq \mu(S_i)$ for all $i$. Then, the expected cost of the game is at most:

$$\mu(S_1) + \sum_{i=2}^{k} \Pr\left(S_i = \text{winner}(K)\right)\mu(S_i). \tag{3}$$

We first provide some intuition for the proof by considering the case when $S_1$ does not intersect with sets $S_2, \ldots, S_k$, i.e. sets $S_1$ and $S_i$ are disjoint for all $i = 2, 3, \ldots, k$. We split all sets into two groups $S_1$ and the rest of the sets $S_2, \ldots, S_k$. We know from Lemma 2.5 that the winner among all sets $S_1, \ldots, S_k$ is either $S_1$ or winner $\left(\{S_2, \ldots, S_k\}\right)$. Denote $I^- = \{S_2, \ldots, S_k\}$. Each set $S_i$ is eliminated at time $e(S_i)$. The set $S_1$ is eliminated at its hitting time $h(S_1)$ unless it is the only remaining set at time $h(S_1)$ (because we are considering the case when $S_1$ does not overlap with other sets). Thus,

$$\text{winner}(K) = \begin{cases} S_1, & \text{if } h(S_1) > e(\text{winner}(I^-)); \\ \text{winner}(I^-), & \text{if } e(\text{winner}(I^-)) > h(S_1). \end{cases} \tag{4}$$

When the winner among $S_1, \ldots, S_k$ is not $S_1$, we consider two cases of the winner $S_i$: (1) $S_i$ is a surprise set; (2) $S_i$ is a non-surprise set.

**Definition 3.1.** *We say that $S_i$ is a surprise set if $e(S_i) \geq h(S_1) \geq L/\mu(S_i)$, where $L = \ln k$.*

We call $S_i$ a surprise set because the probability of the event $e(S_i) \geq h(S_1) \geq L/\mu(S_i)$ is small. We give a bound on the probability of $e(S_i) \geq h(S_1) \geq L/\mu(S_i)$ in Lemma 3.3. Here, we provide some intuition. By Lemma 2.9, the hitting time $h(S_i)$ is an exponential random variable with parameter $\mu(S_i)$. Thus, the expected hitting time for $S_i$ is $1/\mu(S_i)$. Consider a set $S_i$ with a small measure ($\mu(S_i)$ is close to $\mu(S_1)$). If the hitting time $h(S_1) \geq L/\mu(S_i)$, then $h(S_1)$ is much larger than its expected value $1/\mu(S_1)$, which happens with a small probability. Consider a set $S_i$ with a large measure $\mu(S_i) \gg \mu(S_1)$. Then, the expected hitting time for $S_i$ is $1/\mu(S_i)$, which is much smaller than the expected hitting time of $S_1$. Thus, the event $e(S_i) \geq h(S_1)$ occurs with a small probability.

Let us examine bound (3). Let $Surprise$ be the set of all surprise sets. Note that $Surprise$ is a random set. Then,

$$\sum_{i=2}^{k} \Pr\left(S_i = \text{winner}(K)\right)\mu(S_i) \leq \sum_{i=2}^{k} \Pr\left(S_i = \text{winner}(K),\ S_i \notin Surprise\right) \cdot \mu(S_i) \quad (5)$$

$$+ \sum_{i=2}^{k} \Pr\left(S_i \in Surprise\right) \cdot \mu(S_i).$$

We show in the next section (Lemma 3.3) that the second sum is upper bounded by $\mu(S_1)$. We now bound the first sum. For every winner $S_i$ which is not a surprise set, we have $e(S_i) \geq h(S_1)$ (because $S_i$ is the winner) and $h(S_1) \leq L/\mu(S_i)$ (because $S_i$ is not a surprise set). We also have $S_i = \text{winner}(I^-)$, thus

$$\Pr\left(S_i = \text{winner}(K),\ S_i \notin Surprise\right) \leq \Pr\left(h(S_1) \leq L/\mu(S_i) \text{ and } S_i = \text{winner}(I^-)\right).$$

By Lemma 2.9, all hitting times $h(S_i) = \min_{\omega \in S_i} h(\omega)$ for $i \geq 2$ are independent from $h(S_1)$. Thus, $\text{winner}(I^-)$ is also independent of $h(S_1)$ ($\text{winner}(I^-)$ depends only on the hitting times for sets $S_i \in I^-$). Therefore,

$$\Pr\left(S_i = \text{winner}(K),\ S_i \notin Surprise\right) \leq \Pr\left(h(S_1) \leq L/\mu(S_i)\right) \cdot \Pr\left(S_i = \text{winner}(I^-)\right)$$

$$= \underbrace{\left(1 - e^{-L\mu(S_1)/\mu(S_i)}\right)}_{\leq L\mu(S_1)/\mu(S_i)} \cdot \Pr\left(S_i = \text{winner}(I^-)\right)$$

$$\leq \Pr\left(S_i = \text{winner}(I^-)\right) \cdot L \cdot \mu(S_1)/\mu(S_i).$$

We combine all bounds on terms of (5) and get the following bound on the expected cost of the game:

$$\mu(S_1) + \sum_{i=2}^{k} \Pr\left(S_i = \text{winner}(I^-)\right) \cdot L \cdot \mu(S_1) + \mu(S_1) = (L+2) \cdot \mu(S_1) = (\ln k + 2) \cdot \mu(S_1).$$

This concludes the proof of the theorem for the case when $S_1$ does not overlap with $S_2, \ldots, S_k$. We now analyze surprise sets.

## 3.1 Surprise Sets

In this section, we prove a bound on the probability that a set $S_i$ is a surprise set. We no longer assume that $S_1$ does not intersect with other sets $S_i$. We first show a lemma about exponential random variables.

**Lemma 3.2.** *Let $X$ and $Y$ be two independent exponential random variables with positive parameters $\lambda_X$ and $\lambda_Y$, respectively. Then, for every $T \geq 0$, we have*

$$\Pr\left(Y \geq X \geq T\right) = \frac{\lambda_X}{\lambda_X + \lambda_Y} \cdot e^{-(\lambda_X + \lambda_Y)T}. \quad (6)$$

*Proof.* The desired probability can be easily found by computing $\int_T^\infty (F_X(t) - F_X(T))f_Y(t)dt$, where $F_X(t) = 1 - e^{-\lambda_X t}$ is the cumulative distribution function of $X$, and $f_Y(t) = \lambda_Y \cdot e^{-\lambda_Y t}$ is the probability density function of $Y$. Here, we give an alternative proof. Write,

$$\Pr\left(Y \geq X \geq T\right) = \Pr\left(Y \geq X \ \& \ \min(X, Y) \geq T\right)$$

$$= \Pr\left(X \leq Y \mid \min(X, Y) \geq T\right) \cdot \Pr\left(\min(X, Y) \geq T\right).$$

We have $\Pr\left(\min(X, Y) \geq T\right) = e^{-(\lambda_X + \lambda_Y)T}$, because the minimum of two independent exponential random variables with parameters $\lambda_X$ and $\lambda_Y$ is an exponential random variable with parameter $\lambda_X + \lambda_Y$. Then, $\Pr\left(X \leq Y \mid \min(X, Y) \geq T\right) = \Pr\left(X \leq Y\right)$ because the exponential distribution is memoryless; and $\Pr\left(X \leq Y\right) = \lambda_X/(\lambda_X + \lambda_Y)$. $\square$

**Lemma 3.3.** *For every set $S_i$, we have*

$$\Pr(S_i \text{ is surprise set}) \leq \frac{1}{k} \cdot \frac{\mu(S_1)}{\mu(S_i)}.$$

*Proof.* First, we show that $\min(e(S_i), h(S_1)) \leq h(S_i \setminus S_1)$.

**Claim 3.4.** *We always have* $\min(e(S_i), h(S_1)) \leq h(S_i \setminus S_1)$.

*Proof.* Consider an arbitrary realization of the game and the time $t = h(S_i \setminus S_1)$ when $S_i \setminus S_1$ is hit. If by this time, $S_1$ has already been hit then $h(S_1) < t$. Similarly, if by this time, $S_i$ has already been eliminated then $e(S_i) < t$. Otherwise, both $S_1$ and $S_i$ are still remaining in the game at time $t$. Therefore, when we pick $\omega \in S_i \setminus S_1$ at time $t$, set $S_i$ gets eliminated (since $\omega \in S_i$; $\omega \notin S_1$; both $S_1$ and $S_i$ are remaining in the game). Thus, in this case, $e(S_i) = t$. This concludes the proof. $\square$

If $S_i$ is a surprise set, then $\min(e(S_i), h(S_1)) = h(S_1) \geq L/\mu(S_i)$. By Claim 3.4, we have

$$h(S_i \setminus S_1) \geq \min\big(e(S_i), h(S_1)\big) = h(S_1) \geq L/\mu(S_i).$$

Thus, $\Pr(S_i \text{ is surprise set}) \leq \Pr\Big(h(S_i \setminus S_1) \geq h(S_1) \geq L/\mu(S_i)\Big)$. By Lemma 3.2 applied to the independent exponential random variables $h(S_1), h(S_i \setminus S_1)$, and time $T = L/\mu(S_i)$, we have

$$\Pr(S_i \text{ is surprise set}) \leq \frac{\mu(S_1)}{\mu(S_i \setminus S_1) + \mu(S_1)} \cdot e^{-\frac{L(\mu(S_i \setminus S_1) + \mu(S_1))}{\mu(S_i)}} \leq \frac{1}{k} \cdot \frac{\mu(S_1)}{\mu(S_i)}.$$

$\square$

## 3.2 General Case

*Proof of Theorem 2.1.* We upper bound the expected cost of the game for arbitrary sets $S_1, \ldots, S_k$. As before, we assume that $S_1$ is the smallest set. We remind the reader that each hitting time $h(S_i)$ is an exponential random variable with parameter $\mu(S_i)$. In the proof, we will use the definitions of surprise sets (see Definitions 3.1). We also set $L = \ln k$. We define all sets $S_i$ for $i \neq 1$ that are not a surprise set to be non-surprise sets.

We separately upper bound the cost of the winner depending on whether the winner is (a) set $S_1$, (b) surprise set, (c) non-surprise set. Write

$$\mathbf{E}\big[\mu(\mathrm{winner}(K))\big] = \mathbf{E}\big[\mu(\mathrm{winner}(K)) \cdot \mathbf{1}\{\mathrm{winner}(K) = S_1\}\big] \tag{a}$$

$$+ \mathbf{E}\big[\mu(\mathrm{winner}(K)) \cdot \mathbf{1}\{\mathrm{winner} \text{ is surprise set}\}\big] \tag{b}$$

$$+ \mathbf{E}\big[\mu(\mathrm{winner}(K)) \cdot \mathbf{1}\{\mathrm{winner} \text{ is non-surprise set}\}\big]. \tag{c}$$

Term (a) is upper bounded by $\mu(S_1)$. We bound term (b) using Lemma 3.3: The probability that a set is a surprise set is at most $1/k \cdot \mu(S_1)/\mu(S_i)$. Thus, the expected total measure of all sets (not only the surprise winner) is upper bounded by $\frac{1}{k} \sum_{i=2}^{k} \frac{\mu(S_1)}{\mu(S_i)} \mu(S_i) < \mu(S_1)$.

We now bound term (c). Define a new random variable: Let $\mathrm{cost}(\omega)$ be the cost of the winner (i.e., $\mu(S_i)$, where $S_i$ is the winner) if (1) the winner is a non-surprise set, and (2) $\omega$ is the first element that was chosen in $S_1$. We let $\mathrm{cost}(\omega) = 0$, otherwise. If $\omega$ is the first element that was chosen in $S_1$, then $h(S_1) = h(\omega)$. So, the definition of $\mathrm{cost}(\omega)$ can be written as follows:

$$\mathrm{cost}(\omega) = \mu(\mathrm{winner}(K)) \cdot \mathbf{1}\{h(S_1) = h(\omega)\} \cdot \mathbf{1}\{\mathrm{winner}(K) \notin Surprise\}.$$

Since the hitting time $h(S_1)$ is finite with probability 1, the term (c) equals

$$(c) = \sum_{\omega \in S_1} \mathbf{E}[\mathrm{cost}(\omega)].$$

Lemma 3.5, which we prove below, gives a bound of $2L\mu(S_1)$ on the expression above. Combining upper bounds on terms (a), (b), and (c), we get

$$\mathbf{E}\big[\mu(\mathrm{winner}(K))\big] \leq (1 + 2L + 1)\mu(S_1) = (2\ln k + 2) \cdot \mu(S_1).$$

$\square$

**Lemma 3.5.** *For every* $\omega \in S_1$, *we have* $\mathbf{E}[\mathrm{cost}(\omega)] \leq 2L\mu(\omega)$.

*Proof.* We have

$$\mathbf{E}[\text{cost}(\omega)] = \mathbf{E}\Big[\mu(\text{winner}(K)) \cdot \mathbf{1}\{h(S_1) = h(\omega)\} \cdot \mathbf{1}\{\text{winner}(K) \notin Surprise\}\Big]. \quad (7)$$

If $S_i$ is a non-surprise set, then $h(S_1) < L/\mu(S_i)$ or $e(S_i) < h(S_1)$. If $S_i$ is the winner, then $e(S_i) \geq h(S_1)$. Thus, if $S_i$ is a non-surprise winner, then $h(S_1) < L/\mu(S_i)$. This observations gives us the following upper bound on (7):

$$\mathbf{E}\big[\text{cost}(\omega)\big] \leq \sum_{i=2}^{k} \mu(S_i) \cdot \Pr\Big(S_i = \text{winner}(K) \text{ and } h(\omega) = h(S_1) < L/\mu(S_i)\Big). \quad (8)$$

Define two set systems $I_\omega^-$ and $I_\omega^+$ of sets $S_i$ containing and not containing $\omega$:

$$I_\omega^- = \{S_i : \omega \notin S_i \text{ and } i \geq 2\};$$
$$I_\omega^+ = \{S_i : \omega \in S_i \text{ and } i \geq 2\}.$$

Note that $K \equiv \{S_1, \ldots, S_k\} = \{S_1\} \cup I_\omega^- \cup I_\omega^+$. By Lemma 2.5,

$$\text{winner}(K) \in \big\{S_1, \text{winner}(I_\omega^-), \text{winner}(I_\omega^+)\big\}.$$

Observe that if $S_i$ with $i \geq 2$ is the winner, then $S_i = \text{winner}(I_\omega^-)$ or $S_i = \text{winner}(I_\omega^+)$. We replace the condition $S_i = \text{winner}(K)$ with $S_i \in \{\text{winner}(I_\omega^-), \text{winner}(I_\omega^+)\}$ in (8) and get bound:

$$\mathbf{E}\big[\text{cost}(\omega)\big] \leq \sum_{i=2}^{k} \mu(S_i) \cdot \Pr\Big(S_i \in \{\text{winner}(I_\omega^-), \text{winner}(I_\omega^+)\} \text{ and } h(\omega) < \frac{L}{\mu(S_i)}\Big).$$

The key observation now is that sets $\text{winner}(I_\omega^-)$ and $\text{winner}(I_\omega^+)$ are independent of $h(\omega)$. This is the case, because sets remaining in the competitions $\mathcal{R}_n(I_\omega^-)$ and $\mathcal{R}_n(I_\omega^+)$ do not change when we select $\omega$. The set $\mathcal{R}_n(I_\omega^-)$ does not change in the round $n$ when $\omega$ is chosen because all sets $S_i$ in $\mathcal{R}_n(I_\omega^-) \subset I_\omega^-$ do not contain $\omega$. The set $\mathcal{R}_n(I_\omega^+)$ does not change in this round because all sets $S_i$ in $\mathcal{R}_n(I_\omega^+) \subset I_\omega^+$ contain $\omega$ and consequently when $\omega$ is chosen, none of these sets is removed from $\mathcal{R}_n(I_\omega^+)$ (otherwise, $\mathcal{R}_n(I_\omega^+)$ would become empty). Thus,

$$\mathbf{E}\big[\text{cost}(\omega)\big] \leq \sum_{i=2}^{k} \mu(S_i) \cdot \Pr\big(S_i \in \{\text{winner}(I_\omega^-), \text{winner}(I_\omega^+)\}\big) \cdot \Pr\Big(h(\omega) < \frac{L}{\mu(S_i)}\Big).$$

Using that $h(\omega)$ is an exponential random variable with parameter $\mu(\omega)$, we get (for every $i$)

$$\mu(S_i) \cdot \Pr\Big(h(\omega) \leq \frac{L}{\mu(S_i)}\Big) = \mu(S_i) \cdot \Big(1 - e^{-L\frac{\mu(\omega)}{\mu(S_i)}}\Big) \leq \mu(S_i) \cdot L\frac{\mu(\omega)}{\mu(S_i)} = \mu(\omega)L.$$

Hence,

$$\mathbf{E}\big[\text{cost}(\omega)\big] \leq \mu(\omega)L \cdot \sum_{i=2}^{k} \Pr\big(S_i \in \{\text{winner}(I_\omega^-), \text{winner}(I_\omega^+)\}\big).$$

The sum on the right hand side is at most 2. Thus, $\mathbf{E}[\text{cost}(\omega)] \leq 2L\mu(\omega)$. $\qquad\square$

## Acknowledgments and Disclosure of Funding

The authors are supported by NSF Awards CCF-1955351, CCF-1934931, EECS-29 2216970.

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

# A  Proof of Lemma 2.5

**Lemma 2.5.** *Consider a partitioning of the set system $K = \{S_1, \ldots, S_k\}$ into $m$ sets $I_1, \ldots, I_m$. Then,*
$$\mathrm{winner}(K) \in \big\{ \mathrm{winner}(I_1), \ldots, \mathrm{winner}(I_m) \big\}.$$

The proof of Lemma 2.5 relies on the following observarion.

**Lemma A.1.** *Let $X$ and $Y$ be two subsets of $K$. If $X \subset Y$, then for every $n$, we always have*
$$\mathcal{R}_n(Y) \cap X = \mathcal{R}_n(X) \quad \text{or} \quad \mathcal{R}_n(Y) \cap X = \varnothing. \tag{9}$$

*Proof.* We prove that (9) holds by induction on $n$. Initially, when $n = 0$, we have $\mathcal{R}_0(X) = X$ and $\mathcal{R}_0(Y) = Y$. Therefore, $\mathcal{R}_0(Y) \cap X = X \cap Y = X = \mathcal{R}_0(X)$. Suppose (9) holds for $n$, we prove that (9) also holds for $n' = n + 1$. If $\mathcal{R}_n(Y) \cap X = \varnothing$, then $\mathcal{R}_n(Y) \cap X$ remains empty for all $n' \geq n$. Therefore, (9) holds for $n + 1$. So, let us assume that $\mathcal{R}_n(Y) \cap X = \mathcal{R}_n(X)$. Consider three cases:

- If $\omega_{n+1}$ belongs to all sets in $\mathcal{R}_n(Y)$, then it also belongs to all sets in $\mathcal{R}_n(X) = \mathcal{R}_n(Y) \cap X$. Thus, in this case, no set is eliminated in $X$ or $Y$. That is, $\mathcal{R}_{n+1}(X) = \mathcal{R}_n(X)$ and $\mathcal{R}_{n+1}(Y) = \mathcal{R}_n(Y)$.

- If $\omega_{n+1}$ belongs to all sets in $\mathcal{R}_n(X)$, but not all sets in $\mathcal{R}_n(Y)$, then, at step $n + 1$, we remove all sets that contain $\omega_{n+1}$ and, particularly, all sets in $\mathcal{R}_n(X)$, from $\mathcal{R}_n(Y)$. Consequently, $\mathcal{R}_{n+1}(Y) \cap X = \varnothing$ .

- If not all sets in $\mathcal{R}_n(X)$ and not all sets in $\mathcal{R}_n(Y)$ contain $\omega_{n+1}$, then we remove exactly the same sets from both $\mathcal{R}_n(X)$ and $\mathcal{R}_n(Y) \cap X$. Namely, we remove sets $S_i \in \mathcal{R}_n(Y)$ that contain $\omega_{n+1}$.

We conclude that (9) holds for $n' = n + 1$. $\qquad\square$

*Proof of Lemma 2.5.* Consider an arbitrary realization of the game $\omega_1, \omega_2, \ldots$. Let $n$ be the round when all sets but the winner are eliminated from the competition i.e., $\mathcal{R}_n$ contains only one set, the winner. Since $K$ is the union of $I_1, \ldots, I_k$, the winner must belong to some $I_j$. Now, by Lemma A.1 for $X = I_j$ and $Y = K$, we have $\mathcal{R}_n(K) \cap I_j = \mathcal{R}_n(I_j)$ or $\mathcal{R}_n(K) \cap I_j = \varnothing$. We know that $\mathcal{R}_n(K) = \{\mathrm{winner}(K)\}$ and $\mathrm{winner}(K) \in I_j$. Thus, $\mathcal{R}_n(K) \cap I_j = \{\mathrm{winner}(K)\} \neq \varnothing$, and
$$\mathcal{R}_n(I_j) = \mathcal{R}_n(K) \cap I_j = \{\mathrm{winner}(K)\}.$$

We conclude that at round $n$, $\mathcal{R}_n(I_j)$ contains only one set – the winner in $K$. Consequently, it is also the winner in $I_j$ i.e., $\mathrm{winner}(I_j) = \mathrm{winner}(K)$. This finishes the proof. $\qquad\square$

