# OpenReview forum: "Random Cuts are Optimal for Explainable k-Medians"
_NeurIPS.cc/2023/Conference — NeurIPS 2023 oral_

### Official Review · Reviewer_vAmy · 2023-07-02

**Soundness:** 4 excellent
**Presentation:** 3 good
**Contribution:** 4 excellent
**Rating:** 8
**Confidence:** 4

**Summary:**

The paper studies the optimal upper bound of explainable k-median and shows a tight analysis for the random coordinate cut algorithm. The explainable k-clustering model was first introduced by Dasgupta et al. [ICML’20] and was followed by a flurry of work to understand the optimal approximation factor to achieve explainability. For the k-median task, previous work has shown that an $\Omega(\log k)$ factor is necessary, and algorithms that follow a random coordinate cut scheme can achieve $O(\log k \cdot \log\log k)$-approximation. This paper shows that by picking appropriate distributions, the random coordinate cut can achieve the asymptotically tight approximation of $O(\log k)$.

At the heart of the analysis of the paper is a newly-defined *set elimination game* (SEG), where elements are split into multiple (potentially overlapping) sets, and the referee takes an element $e$ at random and eliminates all the sets that contain $e$ unless every remaining set contain $e$. The final value of the game is defined as the measure of the final surviving set. The paper establishes the upper bound for explainable k-median clustering by: 1). showing a ‘’reduction’’ from the cost of explainable k-median of a single point $x$ to the value of the SEG, 2). proving the expected value of the game as $O(\log k)\cdot \mu(S^*)$, where $S^*$ is the set with the smallest measure.

In my opinion, the paper is generally well-written, and it settles an important question in explainable k-median clustering. The proof idea is novel, and it is presented in a clean manner (minus some minor exposition issues, see the weakness for details). Therefore, I believe the merits of the paper are sufficient for acceptance.

**Strengths:**

Explainable clustering has been a popular topic in the recent few years, and the paper settles an open problem in the area. The idea of set elimination games is novel and interesting, and the paper is well-written. The organization helps the reader a lot in understanding the paper.


**Weaknesses:**

I do not see any major technical weaknesses in the paper. On the exposition part, I have some minor comments:
- The paper is written in relatively advanced technical languages, and they might not be readily accessible to the broader NeurIPS audiences. To make the readers’ task easier, I think it makes sense to add some more intuitions on why using the set elimination game, why the value should be small, and the high-level intuitions for the analysis.
- Conversely, the paper spent little effort discussing why explainability in k-clustering is important and why shaving off the exponentially-small $\log\log k$ factor is significant. I would recommend adding some discussion on this front for later versions.
- Some notation problems. In the proof of Theorem 2.2, there is a slight overload of notation in the construction of $\Omega$, and $S$: $S$ was used as a *variable* for the sets in the definition of the SEG, but in the proof it becomes the *realization*, i.e., fixed sets. I would recommend using a different notation instead. Also, $x_j$ (which I guess means the j-th coordinate of the point $x$) is used before it is defined.


**Questions:**

- You never mentioned the initialization of the reference sets, but judging from the proof of Theorem 2.2, you’re assuming the optimal k-median centers as the reference centers, right? Something like this should be mentioned and discussed.
- You never mentioned the *distribution* of the random coordinate cut in the paper; from the requirement of Theorem 2.1, it seems uniform at random works. Is this right? You should explicitly define the distribution, either way (‘’pick at random’’ $\neq$ ‘’pick uniformly at random’’ – the former only means picking something from *a* distribution).
- What is the intuition for the 'surprising sets'? I can read that the name 'surprising' may come from the fact that for $h(S_1) \cdot \mu(S_i) \geq \log k$, either $S_1$ should be hit very late or $S_i$ should have a high volume, which means it should have been hit/eliminated way earlier. If this is true, please also add a discussion about it, as it'll help readers understand your proof.
- Do you see any other applications for the Set Elimination Game?


**Limitations:**

The work is of theoretical nature, so the (non-technical) limitations are on the model side, e.g., the explainable clustering model might not provide easy-to-understand explanations in real life. However, I do not think this affects the significance of the contribution.

---

> ### Author Rebuttal · Authors · 2023-08-07
>
> We thank the reviewer for suggesting how we can improve the presentation. We will address all your comments in the final version of the paper.
>
> 1. For Theorem 2.2, we do not assume the reference centers are optimal centers for k-medians. We show that for every set of k reference centers, the Random Coordinate Cut finds a threshold tree with the cost at most $2\ln k+2$ times the unconstrained cost given by the input reference centers. We define the competitive ratio as the ratio between the cost of the decision tree and the cost of the input reference centers. The price of the explainability for k-medians is the ratio of the cost of the decision tree and the optimal unconstrained k-medians cost. To get an $O(\log k)$ upper bound on the price of explainability, we need to use a constant factor approximation algorithm for k-medians to find the reference centers.
>
> 2. Yes. Threshold cuts are sampled uniformly at random.
>
> 3. Yes. We will add a discussion about the surprise sets.
>
> 4. The Set Elimination Game can potentially be used to analyze other weighted sampling algorithms with non-independent events but we are not aware of any specific examples now.

---

> > ### Comment · Reviewer_vAmy · 2023-08-11
> >
> > Thanks for the responses. I think it is worth adding more discussions according to your answers, especially for point 1.

---

### Official Review · Reviewer_uidA · 2023-07-03

**Soundness:** 4 excellent
**Presentation:** 4 excellent
**Contribution:** 4 excellent
**Rating:** 8
**Confidence:** 4

**Summary:**

The paper provides a tight analysis of the random-coordinate-cut (RCC shortly) algorithm for explainable k-medians in the $ \ell_1 $ metric. To be precise, the RCC algorithm was independently proposed  by several works, and for the explainable k-medians in $ \ell_1 $ , the tightest proved competitive ratio is $ O(\log k \log\log k) $, which has a small gap between its lower bound $ \Omega(\log k) $. This paper shows that the competitive ratio is no larger than $ 2\ln k+2 $, thereby concluding that the RCC algorithm is optimal for explainable k-medians.

**Strengths:**

- This paper completely answers the problem of whether there exists an algorithm with a competitive ratio that matches the lower bound $ \Omega(\log k) $ for the explainable k-medians problem.  Its contribution is significant and solid. The proofs are simple and elegant.
- The idea behind the analysis method set elimination game is ingenious. Actually, it effectively summarizes a series of prior works in a concise and abstract manner.

**Weaknesses:**

i don’t find obvious weaknesses

**Questions:**

- I have no questions but the independent work of this paper. I am interested in learning more about the comparison between these two works, particularly with regard to the differences in the techniques and high-level ideas underlying them.

---

### Official Review · Reviewer_9ZE5 · 2023-07-06

**Soundness:** 4 excellent
**Presentation:** 4 excellent
**Contribution:** 4 excellent
**Rating:** 8
**Confidence:** 4

**Summary:**

The paper considers the problem of explainable $k$-medians which was recently introduced by Dasgupta et al, and analyzes the RandomCoordinateCut algorithm. The paper proves an upper bound of $2\ln k + 2$ on the competitive ratio of RandomCoordinateCut. This matches a previously known lower bound of $\Omega(\log k)$, and hence the paper proves that RandomCoordinateCut for explainable $k$-medians is optimal up to constant factors.

Edit: I've read the authors' rebuttals. As I mentioned in my first review, I believe that this is a strong accept paper.

**Strengths:**

The paper proves optimality of RandomCoordinateCut for $k$-medians with a remarkably simple and elegant proof. Furthermore, the paper is generally very well written and easy to follow.

**Weaknesses:**

I can't see any significant weakness in the paper.

Here are few minor comments/typos:
- Page 4, line 116: The equation $Pr(\omega_n = \omega) = \mu(\omega)/\mu(\Omega)$  is not strictly formal for spaces that are not discrete.
- Page 5: In the displayed equation right after line 156, the summation should be over $j$.
- Page 5, line 185: "all sets are non-empty and disjoint" -> "all sets are non-empty and distinct".
- Page 8: In the displayed equation after line 296, I think that in (c) we should also specify that the winner is not $S_1$, because technically $S_1$ does not satisfy the definition of a surprise set.

**Questions:**

- Can the authors comment about the proof of Gupta et al. (mentioned on page 4, line 105)? Is it very different from the approach that is presented in the current paper?

**Limitations:**

No concerns regarding potential societal impact of this work.

---

### Official Review · Reviewer_jrDD · 2023-07-07

**Soundness:** 4 excellent
**Presentation:** 4 excellent
**Contribution:** 4 excellent
**Rating:** 8
**Confidence:** 3

**Summary:**

This paper considers the Explainable k-Medians problem, which is a more interpretable model for k-clustering than traditional models such as k-median. An explainable clustering partitions space based on coordinate threshold cuts, so that instead of a Voronoi diagram as in k-median / k-means, we end up with rectangular partitions. More specifically, the partitions are determined by a binary decision tree where each node contains a coordinate and a threshold, and the left child corresponds to points that fall below the threshold in that coordinate, and the right child corresponds to points that fall above the threshold in that coordinate. An algorithm is f(k)-competitive for explainable k-median if it produces an explainable clustering that has cost at most f(k) times the cost of the optimal (unconstrainted) k-median clustering. This paper shows that Random Coordinate Cut algorithm (studied by several authors before this paper) has the optimal competitive ratio: specifically, they prove it is (2 \log k + 2)-competitive--improving on the previously best known bound of O(log log k)--and matching a previously known \Omega(log k) lower bound.

The technique is to reduce the problem to a game which the authors call the Set Elimination Game, which is syntactically similar to the Random Coordinate Cut algorithm (and indeed, the authors note that other authors have implicitly analyzed this game). In it, there is some ground set \Omega and an associated probability measure \mu, along with a set of subsets. At each round, an element of \Omega is chosen according to \mu, and any remaining subset that contains this element is eliminated (unless all remaining subsets contain this element). The last remaining subset is the winner, and the cost of the game is \mu(winner). The main result, which directly translates to the competitive ratio bound for explainable k-median, is that the expected cost of the game is at most (2 log k + 2) times the cost of any subset.

The idea behind the analysis is to associate each round with an arrival in a Poisson process, so that they may then view the times at which elements first are chosen in the game as hitting times with exponential distributions.  The hitting times have the nice property that they are independent for different elements. Notably, hitting times single-handedly determine the behavior of the game, so the analysis of the game can be reduced to the analysis of hitting times.

**Strengths:**

- This paper resolves the question of whether the Random Coordinate Cut algorithm is optimal via a tight analysis.

- The analysis of the Set Elimination Game using Poisson processes is creative and novel (based on my understanding from the related work section, other papers implicitly analyze this game, but the method here is new). It is also quite clean.

- The formulation of the Set Elimination Game and the main result on its expected cost (Theorem 2.1) may be of independent interest.

**Weaknesses:**

The result is slightly weaker (by a factor of 2) than a concurrent result of Gupta, Pittu, Svensson, and Yuan (2023), which gives a bound of 1+H_{k-1}.

**Questions:**

Can any of the methods in this paper potentially be adapted to explainable k-means?

**Limitations:**

Not applicable.

---

> ### Author Rebuttal · Authors · 2023-08-07
>
> Yes, our result can be used for explainable k-means. Makarychev and Shan (2021) provided a cut-preserving terminal embedding from $\ell_2^2$ to $\ell_1$ with distortion $O(k)$. Thus, we can first run the terminal embedding and then use the Random Coordinate Cut algorithm on the new instance. Our analysis and also the analysis by Gupta et al. (2023) show that this algorithm achieves $O(k\log k)$ competitive ratio for explainable k-means. This ratio matches the previous best ratio given by Esfandiari, Mirrokni, and Narayanan (2022). Gupta et al. (2023) proposed a new algorithm that achieves the $O(k \log\log k)$ competitive ratio for k-means.

---

> > ### Comment · Reviewer_jrDD · 2023-08-18
> >
> > I thank the authors for their response and maintain my evaluation.

---

### Author Rebuttal · Authors · 2023-08-07

First of all, we would like to thank all the reviewers for their valuable feedback. We will address
all their comments and implement all their suggestions in the final version of the paper.

Several reviewers asked us to compare our proof and the proofs obtained independently by Gupta, Pittu, Svensson, and Yuan (2023). Gupta et al (2023) gave two different proofs. In the first, simpler proof, they also used the exponential clock technique. However, this proof is somewhat different from ours. First, it does not consider the set elimination game and instead works directly with cut metrics. Second, it keeps track of the elimination time of set $S_1$ (using our terminology) rather than its hitting time. This makes the technical details a bit different from ours. Their proof gives a slightly better bound than ours (their bound is $(1+o(1))\ln(k)$, our bound is $2\ln k +2$). The second more involved proof provided by Gupta, Pittu, Svensson, and Yuan (2023) gives a tight competitive ratio of $1+H_{k-1}$. This proof uses a different approach: It reduces an arbitrary instance of the problem to the “uniform” instance, in which all centers other than the closest center to point x are at the same distance from x.

---

### Decision · Program_Chairs · 2023-09-21

**Decision:**

Accept (oral)

**Comment:**

The reviewers greatly appreciated the main contribution of the paper, and there was consensus that this a very strong submission.